# The Impact of Birth Season and Sex on Motor Skills in 2-Year-Old Children: A Study in Jinhua, Eastern China

**DOI:** 10.3390/life14070836

**Published:** 2024-06-30

**Authors:** Yuanye Zhu, Shuying Wang, Yongdong Qian, Jiahui Hu, Huiling Zhou, Mallikarjuna Korivi, Weibing Ye, Rong Zhu

**Affiliations:** 1Institute of Human Movement and Sports Engineering, College of Physical Education and Health Sciences, Zhejiang Normal University, Jinhua 321004, China; zhuyuanye@zjnu.edu.cn (Y.Z.); tyxyqyd@zjnu.cn (Y.Q.); hujiahui@zjnu.edu.cn (J.H.); huiling@zjnu.edu.cn (H.Z.); mallik.k5@gmail.com (M.K.); 2College of Physical Education, North University of China, Taiyuan 030051, China; 2111014205@st.nuc.edu.cn; 3Department of Sports Science, Wenzhou Medical University, Wenzhou 325035, China

**Keywords:** children, seasonal effect, motor development, PDMS-2, gross motor, fine motor

## Abstract

Background: This study investigates the effects of birth season and sex on the development of gross and fine motor skills in 2-year-old children in Jinhua, Eastern China. Methods: Conducted in Jinhua, a city in central Zhejiang Province, Eastern China, this research involved 225 children, assessing their gross and fine motor skills using the Peabody Developmental Motor Scales, Second Edition. Scores were adjusted for age in months to avoid the relative age effect. Statistical analyses included MANOVA to evaluate the impacts of season and sex. Results: Sex had no significant impact on overall motor development scores (*p* > 0.05). However, the season of birth significantly affected fine motor quotient (FMQ) and total motor quotient (TMQ) (*p* < 0.05). Boys’ motor skills were generally unaffected by season, whereas girls born in winter exhibited superior fine motor skills compared to those born in summer. Conclusions: Seasonal environmental factors significantly influence early motor development, particularly fine motor skills in girls. These findings highlight the importance of considering seasonal variations in early childhood interventions aimed at enhancing exercise physiology and sports performance.

## 1. Introduction

The 270-day gestation period and the subsequent two years (730 days) of life together form an essential and sensitive period for human development [1,2]. Evidence suggests that high-quality early childhood development can significantly enhance academic outcomes [3], improve quality of life [4], bolster executive functioning [5], and even contribute to reductions in societal crime rates [6] and increased societal earnings [7]. Conversely, children with developmental challenges often exhibit academic underperformance and are more prone to conditions like depression [8], anxiety [8], and obesity [9]. Numerous factors influence early childhood development, including the home environment [10], caregiver response style [11], neighborhood built environment [12], and feeding type [13]. 

Given the sensitivity to environmental factors during early life, seasonal factors such as temperature variations, the availability of fresh vegetables, the prevalence of diseases, and the duration of sunlight exposure can all impact infants’ growth [14]. Prior research has linked birth season to various developmental outcomes, such as emotional and behavioral regulation [15] and childhood stunting [16]. For example, in rural northwest China, infants born in winter have been found to have higher hemoglobin (Hb) concentrations compared to those born in summer. Similarly, cognitive development scores and psychomotor development scores were significantly higher among winter-born infants [17].

Motor skills, which refer to the nervous system’s ability to control and coordinate movements, can be broadly categorized into gross and fine motor abilities [18]. The development of these skills, which underscores a child’s capacity to engage with their surroundings, is paramount during early childhood [19]. While numerous studies have delved into the relationship between birth season and motor development, the findings have been inconclusive. Some research suggests that birth season can influence young children’s motor skills [14,20,21], while a study by Gil [22] found no such association. A 3-year follow-up study of 23 healthy infants who received 400 IU/d of supplemental vitamin D found that no significant effects with respect to the season at birth were observed in the gross motor quotient (GMQ) [23]. Yasumitsu-Lovell, K. et al. [14] found that summer-born Japanese infants had the worst outcomes at both 6 months and 12 months of age. Tsuchiya, K.J. et al. [24] found that among Japanese infants aged 6 months, those born in spring showed better neuromotor development compared to those born in autumn. However, by 14 months, these differences disappeared. Notably, the majority of these studies predominantly focus on gross motor skills, often overlooking the potential relationship between birth season and children’s fine motor development.

Scientific evidence indicates that sex can influence motor competence. For instance, boys often demonstrate greater proficiency in gross motor skills, whereas girls tend to excel in fine motor skills, such as drawing and writing [25]. Additionally, the trimester of birth may affect motor competence due to the relative age effect, where older children within the same age group exhibit more advanced motor skills than their younger peers [26,27]. Considering the presence of the relative age effect, it is important to evaluate motor skills using month-standardized scores. Furthermore, within the same class, younger children may experience the “Matthew effect,” where children who initially lag in skills due to being younger miss out on opportunities to practice and improve, thus falling further behind over time. To address these factors, we chose to test 2-year-old children who have just entered kindergarten. This allows us to assess their motor skills before the potential impact of formal preschool settings.

Standardized tests to measure motor competence include a variety of tools such as the Alberta Infant Motor Scale, the Gesell Developmental Schedules, the Bruininks-Oseretsky Test of Motor Proficiency (BOT-2), the Movement Assessment Battery for Children (MABC-2), and the Peabody Developmental Motor Scales, Second Edition (PDMS-2) [28]. The Alberta Infant Motor Scale is often used for assessing motor skills in children with cerebral palsy, making it unsuitable for our study’s population. The Gesell Developmental Schedules is now rarely used. While the BOT-2 and MABC-2 are widely recognized and assess both gross and fine motor skills, they have limitations in comprehensively assessing motor skills in 2-year-old children. Additionally, the Bayley Scales of Infant Development-III has not yet been applied in China. The PDMS-2 was chosen for this study because it comprehensively assesses both gross and fine motor skills and has been validated for use in 2-year-old children [29].

In the present study, our objective was to assess the development of both gross and fine motor skills in 2-year-old children. We sought to discern the potential variations in these skills based on the season of birth and further aimed to elucidate any sex-specific influences on motor skills relative to the birth season. To guide our research, we posed the following questions: Are there differences between 2-year-old boys and girls in motor skills depending on the season of birth? What impact does the season of birth have on the various motor skill subtests at this age? By addressing these questions, we aim to contribute to a deeper understanding of the factors influencing early motor skill development.

## 2. Materials and Methods

### 2.1. Recruitment Strategy and Participants 

This study, adopting a cross-sectional design, was conducted in Jinhua, a city in Eastern China’s Zhejiang Province with an approximate population of 1 million. We initially recruited 228 healthy children from two childcare institutions and a kindergarten affiliated with Zhejiang Normal University using a random cluster sampling technique. The study spanned from 18 October 2022 to 20 December 2022. Over this period, 3 children were excluded due to age constraints, culminating in a final sample size of 225 children.

Child participants’ fundamental attributes, such as age, birth date, and sex, were documented at their respective institutions. The cohort comprised 114 boys and 111 girls. The season of birth for each child was deduced from their birth dates. The details of this study were provided with the Chinese version of the written informed consent form. The Chinese version of the informed consent form was signed by the parents or guardians of the children. We also verbally explained the details of this study to the directors of the kindergarten, teachers, parents or grandparents, and students before their voluntary participation. The research design and all associated evaluation procedures received approval from the Institutional Ethical Committee of Zhejiang Normal University under approval number ZSRT2022097.

### 2.2. Assessment of Children’s Basic Characteristics 

Children’s demographic details, specifically sex and birth date, were sourced from their teachers and subsequently recorded for analysis. The season of birth was determined based on each child’s birth date. According to astronomical division, spring (21 March–20 June) begins with the vernal equinox and ends with the summer solstice. Summer (21 June–22 September) begins with the summer solstice and ends with the autumn equinox. Autumn (23 September–20 December) begins with the autumn equinox and ends with the winter solstice. Winter (21 December–20 March) begins at the winter solstice and ends at the spring equinox. 

### 2.3. Evaluation of Children’s Motor Skills 

The motor proficiency of the 2-year-old participants was gauged using the Peabody Developmental Motor Scales, Second Edition [30]. The PDMS-2 encompasses two primary skill categories: gross and fine motor skills. Gross motor skills are further divided into 8 reflection skill (Re:) tasks (0–11 months), 30 stationary skill (St) tasks, 89 locomotion skill (Lo) tasks, and 24 object manipulation skill (Ob) tasks (12–72 months). Fine motor skills comprise 24 grasping skill (Gr) tasks and 26 visual–motor integration skill (Vi) tasks. Each task’s evaluation score spans three levels: 2 points for correct execution, 1 point for partial completion, and 0 points for non-compliance with developmental criteria [30]. The cumulative points constitute the raw scores for each skill, which are then converted to standard scores, gross motor quotient (GMQ), fine motor quotient (FMQ), and total motor quotient (TMQ) using tables in the PDMS-2 manual. Importantly, these standard scores are adjusted for the child’s age in months, avoiding the relative age effect. The evaluations took place in vacant halls within the kindergartens and professional training centers during regular hours. The rooms used for evaluation were well lit and quiet, minimizing distractions. Children were dressed in their regular attire, which typically included comfortable, movement-friendly clothing. 

The PDMS-2, employed in this research, stands as a credible and reliable tool for gauging motor skills in Chinese children. It boasts commendable test–retest reliability, content and structural validity, criterion validity, and internal consistency, as corroborated by multiple studies [29,31,32]. Fourteen professionals, encompassing four sports science researchers and ten postgraduate testers, facilitated the assessments. Each PDMS-2 assessment session lasted approximately 30 min per child. Dual independent testers concurrently observed each child’s performance, adhering strictly to PDMS-2 guidelines [30]. All requisite equipment was readied ahead of the assessment day. On the day of evaluation, participants received verbal briefings, supplemented with precise demonstrations on skill scoring. The tests were administered at similar times during the day across all participating locations to control for potential diurnal variations in children’s performance.

### 2.4. Statistical Analysis 

The obtained data (Appendix A) were analyzed using Microsoft^®^ Excel^®^ 2019. Descriptive statistics for participant characteristics were presented as n (%). For the analysis, children were categorized based on sex (two groups) and season of birth (four groups). To appropriately assess the influence of both sex and season of birth on motor development, two separate two-factor multivariate analyses of variance (MANOVA) were conducted. The first MANOVA assessed the effects on GMQ, FMQ, and TMQ, while the second MANOVA focused on the scores of St, Lo, Ob, Gr, Vi, and other subskills. This separation accounts for the differences in the magnitude of these measures. The Bonferroni correction was applied to control for multiple comparisons, and eta squared (η^2^) was calculated to determine the effect size and statistical power of the results. The level of significance was set at *p* < 0.05. This approach ensures that the potential interactions between sex and season of birth are appropriately considered in the analysis, providing a more accurate assessment of their effects on motor development.

## 3. Results

### 3.1. Characteristics of Children

In this study, a total of 225 children aged 2 years participated, of which 114 (50.67%) were boys and 111 (49.33%) were girls. The number of children born in each season was 19 boys and 11 girls born in spring, 42 boys and 32 girls born in summer, 32 boys and 34 girls born in autumn, and 21 boys and 34 girls born in winter (Table 1). 

### 3.2. Effects of Sex and Season of Birth on GMQ, FMQ, and TMQ 

We first conducted a MANOVA on the GMQ, FMQ, and TMQ because these comprehensive measures capture both gross and fine motor development, providing a holistic view of a child’s motor skills. The analysis revealed that the intercept was highly significant (*p* < 0.001, η^2^ = 0.999), indicating that almost all variations in motor development scores are explained by the model. Sex had no significant impact on motor development scores (*p* = 0.106, η^2^ = 0.028), indicating minimal influence. However, the season of birth had a significant effect on motor development scores (*p* = 0.017, η^2^ = 0.030), with a modest but meaningful impact. The interaction between sex and season of birth was not significant (*p* = 0.735, η^2^ = 0.009), suggesting the minimal combined influence of these factors on motor development. These results highlight that while the season of birth significantly affects motor development, sex alone and its interaction with the season do not.

We examined the individual effects on each motor skill measure (Table 2). There were no significant effects of sex on the GMQ, FMQ, and TMQ, indicating similar motor development scores between boys and girls. Significant effects of the season of birth were found on the FMQ and TMQ (*p* < 0.05), suggesting that motor development scores vary with the season of birth. No significant interaction effects were observed between sex and the season of birth, indicating consistent seasonal influences across sexes. These findings highlight that while the season of birth significantly impacts fine and total motor skills, sex and its interaction with the season do not.

### 3.3. Effects of Sex and Season of Birth on Gross and Fine Motor Subskills 

Further analyses of individual motor subskills (stationary, locomotion, object manipulation, grasping, and visual–motor integration) revealed that sex significantly impacts motor skills (*p* < 0.001, η^2^ = 0.131). The season of birth showed a modest impact on these subskills, with some tests indicating significance (*p* = 0.088, η^2^ = 0.034). However, no significant interaction effects were observed between sex and season of birth (*p* > 0.7, η^2^ = 0.017), indicating that the influence of the season of birth on motor skills is consistent across sexes. These results suggest that while sex significantly affects motor skills, the impact of seasonal factors is relatively modest.

The subskills reveal that sex significantly affects locomotion and object manipulation skills (Table 2), while the season of birth significantly impacts grasping skills and may have a modest effect on visual–motor integration skills. No significant interaction effects were observed, indicating that the influence of the season of birth on motor skills is consistent across the sexes.

### 3.4. The Motor Skills Scores of Girls Born in Winter Compared to Those Born in the Summer

Figure 1 shows the influence of the season of birth on the development of motor skills in boys and girls. Initially, we noticed that the motor skills scores (St, Lo, Ob, Gr, and Vi) and motor developmental quotient (GMQ, FMQ, and TMQ) of boys were not significant influenced by the season of birth. We also found that there was no significant difference in the season of birth difference on the St, Lo, Ob, and Vi standard scores and the GMQ and TMQ of girls. However, it is worth noting that girls born in winter showed superior Gr and FMQ (*p* < 0.01) than girls born in summer (Figure 1C,D).

## 4. Discussion

To our knowledge, this study represents a pioneering effort to discern the potential influence of birth season on the motor skills of 2-year-old children. We delved into the motor skill nuances among 2-year-old children from Zhejiang Province, China, juxtaposing the variations against their sex and birth season. Our results underscore that a child’s birth season does indeed sway their motor skill development, and intriguingly, the extent of this effect varies between boys and girls. Specifically, the birth season markedly impacted the fine motor skills of girls, whereas the effect of birth season on motor skills of boys was not significant. Notably, girls born in winter exhibited superior grasping skills compared to their summer-born counterparts. 

The sex-based differentiation in the motor skills of 2-year-old children offers a compelling avenue for exploration. Our results show no significant statistical differences in the Gr, Vi, and FMQ scores among two-year-olds. Similarly to our findings, a study on Saudi Arabian children found no significant differences in fine motor skills between boys and girls [33]. However, studies from Greece [34], Iran [35], and India [36] confirmed that girls tend to outperform boys in fine motor skills. The discrepancies in findings across different studies may be attributed to the varying countries where these studies were conducted [37]. Additionally, these differences might also be due to the distinct characteristics exhibited by different age groups [38].

Our analysis also highlighted that boys scored higher in Ob skills than girls. This aligns with findings from China [39], USA [40], and Spain [41], which highlighted boys’ superior ball skills [42]. Differences in proficiency in object control skills between boys and girls seem to be influenced by evolutionary factors. Early humans lived by throwing stones and swinging clubs. Women invested more resources into reproduction, and men were more likely to be hunters and warriors. These kinds of patterns are inherited through natural selection [43]. Gender differences among children in learning motor skills depend on their family, elder siblings, peers, and teachers with respect to socialization and imitation, and consequently, they participate in activities that fit these gender norms [44]. The better ball skills of boys in our study may also be explained by boys spending more time practicing object-control-related activities, like ball games, thereby developing their ball skills. Additionally, sociocultural factors may also influence the development of children’s motor skills [25].

Our results highlighted that the motor skills of children born in winter significantly surpassed those of summer-born children. This observation aligns with findings from Israel [45], China [17], and the USA [46], all of which underscored the motor developmental edge in winter-born children. Specifically, our data revealed that winter-born girls manifested superior fine motor skills (grasping) compared to summer-born girls, while no such seasonal disparity was evident in gross motor skills. For boys, the birth season did not seem to influence either gross or fine motor skills.

One study suggests that differences in motor skills across seasons are due to the relative age effect (RAE) [26]. However, our research differs from this study in several ways. Firstly, the other study involved older children, such as 4-year-olds, who may have already spent a year or more in kindergarten, thereby being more influenced by group activities and formal education settings. Secondly, the seasonal divisions in their study differ from ours, as their quarters were defined as follows: quarter 1 (q1: born from January to March); quarter 2 (q2: born from April to June); quarter 3 (q3: born from July to September); and quarter 4 (q4: born from October to December). Additionally, their measurement methods may use standard scores based on annual or semi-annual divisions, whereas we calculate standard scores on a monthly basis. This monthly calculation provides a more precise assessment of each child’s development relative to their exact age.

We postulate that the ambient temperature during winter might be a pivotal factor driving superior fine motor development in children. Winter’s chill necessitates bundling up children in layered clothing. Research has indicated that such bulky attire can restrict movement, potentially impeding the exploration and execution of motor skills, predominantly gross motor skills [47]. This scenario might inadvertently provide children with augmented opportunities to hone their fine motor skills. The Peabody Developmental Motor Scales posits that rapid developmental milestones like hand grasping and head movement in infants aged 0-2 months fall under the fine motor domain [30]. Thus, winter’s cold might exert a minimal impact on gross motor skills but accentuate fine motor skill development. Additionally, winter’s shorter days curtail outdoor activities, leading to increased parent–child interactions, which might further amplify this effect. Winter-born infants experiencing relatively fewer outdoor activities may engage more in hand-related activities at home or indoors. Additionally, in summer, environmental factors like temperature might significantly increase the likelihood of premature birth [48], and premature infants tend to lag in motor development compared to full-term infants, which might explain why winter-born infants exhibit better motor skills than those born in summer [49]. The sex-based variations in motor skills across birth seasons might be tethered to parental play promotions. Research indicates that parental play promotions play a pivotal role in optimizing development at six to nine months of age. Parents of boys tend to promote gross motor skills, while those of girls lean towards fine motor skills [25,50]. 

While our study sheds light on intriguing aspects of motor skills development in relation to birth season, it is not without limitations. Firstly, the sample size, encompassing 225 children, is relatively modest. Subsequent research with a more expansive sample is recommended to validate and expand upon our findings. Secondly, the participants in our study hail exclusively from Zhejiang Province, China, characterized by a subtropical monsoon climate. This geographic and climatic specificity may limit the generalizability of our findings to children from different climatic regions. Lastly, our investigation was confined to discerning the impact of birth season on the motor skills of 2-year-old children. The influence of birth season on motor skills across different age groups, especially those younger than a year, remains an open avenue for exploration.

## 5. Conclusions

Our research highlights a strong association between the season of birth and motor skills development in 2-year-old children. Notably, the fine motor skills, especially grasping, of girls born in winter surpassed those of summer-born girls. This suggests that targeted interventions and motivational strategies might be beneficial for 2-year-old girls born in summer to ensure the balanced development of both gross and fine motor skills during their formative years.

## Figures and Tables

**Figure 1 life-14-00836-f001:**
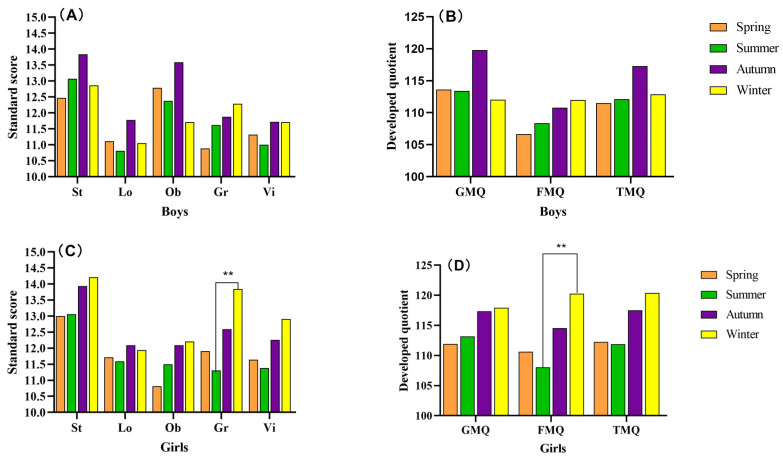
Changes in motor skills scores with different seasons of birth. (**A**), The standard score for boys; (**B**), The developed quotient for boys; (**C**), The standard score for girls; (**D**), The developed quotient for girls; St, stationary skills; Lo, locomotion skills; Ob, object manipulation skills; Gr, grasping skills; Vi, visual–motor integration skills; GMQ, gross motor quotient; FMQ, fine motor quotient; TMQ, total motor quotient. The scores were significant at ** *p* < 0.01 among spring, summer, autumn, and winter.

**Table 1 life-14-00836-t001:** Basic characteristics of the 2-year-old children.

Variables	Boys (*n* = 114)	Girls (*n* = 111)	Total (*n* = 225)
Spring	19 (16.7%)	11 (9.9%)	30 (13.3%)
Summer	42 (36.8%)	32 (28.8%)	74 (32.9%)
Autumn	32 (28.1%)	34 (30.6%)	66 (29.3%)
Winter	21 (18.4%)	34 (30.6%)	55 (24.4%)

Values expressed in numbers (percentage).

**Table 2 life-14-00836-t002:** Seasonal differences in the scores of motor skills for boys and girls.

Variables	GMQ	FMQ	TMQ	St	Lo	Ob	Gr	Vi
Boys (M ± SE)								
Spring	113.63 ± 3.05	106.63 ± 3.20	111.47 ± 3.03	12.47 ± 0.62	11.11 ± 0.51	12.79 ± 0.62	10.89 ± 0.67	11.32 ± 0.55
Summer	113.40 ± 2.05	108.36 ± 2.15	112.14 ± 2.04	13.07 ± 0.42	10.81 ± 0.35	12.38 ± 0.42	11.62 ± 0.45	11.00 ± 0.37
Autumn	119.81 ± 2.35	110.78 ± 2.47	117.28 ± 2.33	13.84 ± 0.48	11.78 ± 0.40	13.59 ± 0.48	11.88 ± 0.52	11.72 ± 0.43
Winter	112.05 ± 2.90	112.00 ± 3.05	112.86 ± 2.88	12.86 ± 0.59	11.05 ± 0.49	11.71 ± 0.59	12.29 ± 0.64	11.71 ± 0.53
Total	114.72 ± 1.31	109.44 ± 1.38	113.44 ± 1.30	13.06 ± 0.27	11.19 ± 0.22	12.62 ± 0.27	11.67 ± 0.29	11.44 ± 0.24
Girls (M ± SE)								
Spring	111.91 ± 4.00	110.64 ± 4.21	112.27 ± 3.98	13.00 ± 0.81	11.73 ± 0.68	10.82 ± 0.82	11.91 ± 0.67	11.64 ± 0.73
Summer	113.16 ± 2.35	108.06 ± 2.47	111.88 ± 2.33	13.06 ± 0.48	11.59 ± 0.40	11.50 ± 0.48	11.31 ± 0.51	11.38 ± 0.43
Autumn	117.32 ± 2.28	114.56 ± 2.39	117.50 ± 2.26	13.94 ± 0.46	12.09 ± 0.39	12.09 ± 0.47	12.59 ± 0.50	12.26 ± 0.41
Winter	117.94 ± 2.28	120.29 ± 2.39	120.41 ± 2.26	14.21 ± 0.46	11.94 ± 0.39	12.21 ± 0.47	13.85 ± 0.50	12.91 ± 0.41
Total	115.08 ± 1.41	113.39 ± 1.48	115.51 ± 1.40	13.55 ± 0.29	11.84 ± 0.24 #	11.65 ± 0.29 #	12.42 ± 0.31	12.05 ± 0.26
Effect								
Sex								
F	0.035	3.799	1.177	1.583	4.018	6.032	3.085	3.027
p	0.852	0.053	0.279	0.21	0.046 *	0.015 *	0.080	0.083
η^2^	0.000	0.017	0.005	0.007	0.018	0.027	0.014	0.014
Season								
F	2.236	3.808	2.717	1.710	1.262	1.78	3.584	2.586
p	0.085	0.011 *	0.046 *	0.166	0.288	0.152	0.015 *	0.054
η^2^	0.030	0.050	0.036	0.023	0.017	0.024	0.047	0.035
Sex × Season								
F	1.075	0.963	1.062	0.752	0.202	1.772	1.102	0.35
p	0.361	0.411	0.366	0.522	0.895	0.153	0.349	0.789
η^2^	0.015	0.013	0.014	0.01	0.003	0.024	0.015	0.005

St, stationary skills; Lo, locomotion skills; Ob, object manipulation skills; GMQ, gross motor quotient; Gr, grasping skills; Vi, visual–motor integration skills; FMQ, fine motor quotient; and TMQ, total motor quotient. The scores were significant at * *p* < 0.05; #, different from boys (*p* < 0.05).

## Data Availability

The primary data and contributions presented in this study are encapsulated within the article. For additional details or inquiries, interested parties are encouraged to reach out to the lead authors.

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
