# Peer review of "The Impact of Birth Season and Sex on Motor Skills in 2-Year-Old Children: A Study in Jinhua, Eastern China"

_life, 2024, doi:10.3390/life14070836_

Round 1

Reviewer 1 Report

Comments and Suggestions for Authors

First of all, I would like to thank you for the opportunity to review this manuscript whose objective was to assess the development of both gross and fine motor skills in 2-year-old children. The topic of motor competence (MC) in early childhood is a topic of special relevance since a high degree of performance in motor skills and a high MC has repercussions when it comes to future sports practice and health.

However, the article presented has a series of limitations that I will list below:

The authors try to establish a relationship between aspects of motor competence and the season of the year in which they are born. This is especially difficult since, for example, the seasons as established by the authors are different in the northern hemisphere than in the southern and therefore it could not be established that the season of the year in which schoolchildren are born influences their motor competence. . However, the authors have taken the date of birth as an independent variable, if it has been studied in this population (3-6 years), along with its effect within children who were born within the same cohort but in different months or quarters of the year. same year (relative age effect). Therefore, the design or choice of contrast variables is not correct in the eyes of this reviewer.

Below I will list some details that could help improve the article presented:

Title: "Season of the year" should be changed by quarter of birth.

Abstract:

Line 20: The p-values ​​must be reported

Line 22: As mentioned above, authors must take into account the effect of relative age. It doesn't have so much to do with the season of birth but with the month of birth. This implies that children born in January of the same year compared to those born in December have a year of practice and maturation.

line 22-23: what the authors write has not been verified in the study

Introduction:

In the introduction, authors should include aspects such as the following: what does scientific evidence say about gender and motor competence?; What does the scientific evidence say about the differences between the trimester of birth and motor competence? What types of standardized tests exist to measure motor competence? of product? of result? Which one is better? Why was the PDMS-2 chosen for this study?

Line 39: authors must include and present the current literature on the trimester of birth and the effect of relative age on motor competence.

line 52-56: This has no basis, the basis is the month of birth. In addition, the authors must ask research questions that can be answered through the results obtained and at the same time can guide the discussion. For example: are there differences between 2-year-old boys and girls in manipulative skills depending on the trimester of birth?...

Methods:

line 107: The authors must explain what the room where the evaluation was done was like, how the children were dressed, who and what training the evaluators who passed the tests had. Were they the same hours at which the tests were administered to all children and in all Kindergartens?

Statistical analysis: The test is not adequate. When performing the T-test and the ANOVA, measurements are made without taking into account the interaction or effect of the other variable. The correct test would be a multivariate analysis with two factors (gender/:boys-girls) and quarter of birth (q1; q2; q3; q4), with the bonferroni statistic and eta squared for statistical power.

Results

In general, the results must be re-registered based on the indicated statistical test and including the quarter of birth variable along with gender.

line 124-145 (table 1): this reviewer considers that the classification should be done by trimester of birth

line 128: Authors must do new statistical tests that take into account gender and quarter of birth. They should not do this analysis separately.

Discussion:

In the eyes of this reviewer, the season of birth has no basis since in the southern hemisphere, the characteristics of the seasons are different from those of the northern hemisphere. In any case, it should be done based on the quarter of birth if there is scientific evidence to support this (relative age effect). The discussion must be rewritten according to the results obtained from the requested statistical analysis.

lines 171-174: This statement cannot be made since the statistical analysis carried out does not allow it to be done.

lines 177-179: In this comparison, according to the statistical test carried out, the season (quarter of birth) has not been taken into account, this cannot be stated.

Line 189: Have the authors considered that this could be due to the type of gender-related activities that boys and girls carry out? you should value this

line 196: Have the authors considered that this could be due to the type of activities associated with gender that boys and girls carry out? They must value this. the possible explanation provided by the authors should be eliminated

line 202-203: What the authors attribute to the season of birth is typical of the effect of relative age that means that children within the same cohort who are born earlier have more time for maturation and the possibility of practice than those who are born later. in no case can it be attributed to the season of birth

Lines 208-210: This last statement cannot be made by the authors. It has not been evaluated in this research whether the relationship they may have with the variables studied

Conclusions

The conclusions must be written based on the required analyzes and the new results.

References:

Of 48 references, 28 are older than 5 years. Authors should use more current studies for their research, especially in the introduction to know the state of the art.

The authors have used references that are not in line with the study, for example references 1 and 2, or reference 6, which are focused on adolescence. These references are not relevant to this study. Neither does reference 18. (adolescents)

References 19 and 44 have no relationship with the population under study (4-6 years and 3-6) years respectively. The study was carried out with 2-year-old children. A year of difference in these ages represents more than a third of their life and this is very significant in motor development.

I hope my comments can help improve your article.

Author Response

Comment1:  First of all, I would like to thank you for the opportunity to review this manuscript whose objective was to assess the development of both gross and fine motor skills in 2-year-old children. The topic of motor competence (MC) in early childhood is a topic of special relevance since a high degree of performance in motor skills and a high MC has repercussions when it comes to future sports practice and health.

However, the article presented has a series of limitations that I will list below:

The authors try to establish a relationship between aspects of motor competence and the season of the year in which they are born. This is especially difficult since, for example, the seasons as established by the authors are different in the northern hemisphere than in the southern and therefore it could not be established that the season of the year in which schoolchildren are born influences their motor competence. . However, the authors have taken the date of birth as an independent variable, if it has been studied in this population (3-6 years), along with its effect within children who were born within the same cohort but in different months or quarters of the year. same year (relative age effect). Therefore, the design or choice of contrast variables is not correct in the eyes of this reviewer.

A1: We appreciate your insights and apologize for any misunderstandings caused by our manuscript. Specifically, we would like to address the following points:

1.We acknowledge that while we stated in the original manuscript (line 92) that we used standard scores to evaluate motor skills, this was not clearly mentioned in the abstract, leading to some confusion. We apologize for this oversight. The use of standard scores allows us to normalize motor skill performance based on the age of the children in months, ensuring that older children do not have an unfair advantage simply due to their month of birth.

2.We maintain that environmental changes associated with different seasons contribute to the variations in motor competence observed in our study. Our approach takes into account the percentile ranks of motor skills among children of the same age in months, ensuring that motor skill requirements increase with age. Therefore, older children are not inherently advantaged over younger ones.

3.Our testing began in October, meaning the oldest children in our sample were born in October and November. According to our method, these children, classified as autumn-born (born between September 23 and December 20), should have an age advantage. However, if maturation time provided a significant advantage, we would expect winter-born children to show more pronounced differences compared to spring-born children. Instead, our results indicated that winter-born children performed better than summer-born children, suggesting that environmental factors associated with seasons do play a significant role.

In response to the reviewer's comments, we have reiterated in the revised manuscript that we used standard scores(LINE135-136,Line264-275), and we have explicitly stated this in the abstract to avoid any further confusion.

Comment 2: Below I will list some details that could help improve the article presented:

Title: "Season of the year" should be changed by quarter of birth.

A2: Thank you for your valuable feedback. While we maintain that seasonal factors significantly influence motor skills development for the reasons previously stated, we appreciate your suggestion and have revised the title to "The Impact of Birth Season and Sex on Motor Skills in 2-Year-Old Chil-dren: A Study in Jinhua, Eastern China."

Abstract:

Comment 3: Line 20: The p-values must be reported

A3: Thank you for your valuable feedback. We have now included the p-values in the results section as suggested.

Comment 4: Line 22: As mentioned above, authors must take into account the effect of relative age. It doesn't have so much to do with the season of birth but with the month of birth. This implies that children born in January of the same year compared to those born in December have a year of practice and maturation.

A4: Thank you for your insightful feedback. We have taken the relative age effect into account by adjusting the motor skill scores based on the child's age in months. This adjustment ensures that the motor skills development is evaluated fairly, considering the month of birth and mitigating the relative age effect.

Comment 5: Line 22-23: what the authors write has not been verified in the study

A5: Thank you for pointing this out, the statements have been corrected to accurately reflect the verified findings.

Introduction:

Comment 6: In the introduction, authors should include aspects such as the following: what does scientific evidence say about gender and motor competence?; What does the scientific evidence say about the differences between the trimester of birth and motor competence? What types of standardized tests exist to measure motor competence? of product? of result? Which one is better? Why was the PDMS-2 chosen for this study?

A6: Thank you for your insightful suggestions. We have expanded the introduction to include a review of scientific evidence on Sex and motor competence (Line 64-66), the impact of the trimester of birth on motor competence(Line 66-75), and various standardized tests available to measure motor competence. We have also provided a rationale for selecting the Peabody Developmental Motor Scales-Second Edition (PDMS-2) for this study(Line 76-87).

Comment 7: Line 39: authors must include and present the current literature on the trimester of birth and the effect of relative age on motor competence.

A7: We appreciate your feedback. We have added current literature on the trimester of birth and the effect of relative age on motor competence in the introduction (Line 66-75).

Comment 8: line 52-56: This has no basis, the basis is the month of birth. In addition, the authors must ask research questions that can be answered through the results obtained and at the same time can guide the discussion. For example: are there differences between 2-year-old boys and girls in manipulative skills depending on the trimester of birth?...

A8: Thank you for your suggestions. We have revised our research questions to be more specific and answerable based on the results obtained (Line91-94).

Methods:

Comment 9: line 107: The authors must explain what the room where the evaluation was done was like, how the children were dressed, who and what training the evaluators who passed the tests had. Were they the same hours at which the tests were administered to all children and in all Kindergartens?

A9: Thank you for your detailed comments. We have added descriptions of the evaluation environment, children's attire during the tests (LINE136-140), the qualifications and training of the evaluators, and the consistency in test administration times (Line 144-152)

Comment 10: Statistical analysis: The test is not adequate. When performing the T-test and the ANOVA, measurements are made without taking into account the interaction or effect of the other variable. The correct test would be a multivariate analysis with two factors (gender/:boys-girls) and quarter of birth (q1; q2; q3; q4), with the bonferroni statistic and eta squared for statistical power.

A10: Thank you for your statistical guidance. We have revised the statistical analysis to use a multivariate analysis with two factors (gender and quarter of birth), incorporating the Bonferroni correction and reporting eta squared for statistical power (Line 156-166).

Results

Comment 11: In general, the results must be re-registered based on the indicated statistical test and including the quarter of birth variable along with gender.

A11: We appreciate your feedback. We have re-analyzed the results using the suggested statistical tests.

Comment 12: line 124-145 (table 1): this reviewer considers that the classification should be done by trimester of birth

A12: Thank you for your suggestion. We believe that the seasonal classification used in our study is more relevant for examining the environmental impacts on motor skills development. Previous research has indicated that seasonal factors such as temperature and daylight exposure can influence motor development. Therefore, we have retained the seasonal classification to better align with our research objectives.

Comment 13: line 128: Authors must do new statistical tests that take into account gender and quarter of birth. They should not do this analysis separately.

A13: Thank you for your guidance. We have performed new statistical tests that simultaneously take into account sex and seasonal factors (Table 2).

Discussion:

Comment 14: In the eyes of this reviewer, the season of birth has no basis since in the southern hemisphere, the characteristics of the seasons are different from those of the northern hemisphere. In any case, it should be done based on the quarter of birth if there is scientific evidence to support this (relative age effect). The discussion must be rewritten according to the results obtained from the requested statistical analysis.

A14: Thank you for your insightful comments. We have revised the statistical analysis as requested and included sex and season. However, our findings still indicate a significant influence of seasonal factors on motor skills development. While we acknowledge that the characteristics of seasons differ between the northern and southern hemispheres, our study specifically focuses on children in the northern hemisphere. The seasonal classification aligns with existing literature indicating that environmental factors, such as temperature and daylight exposure, play a critical role in motor development. We have updated the discussion to reflect the 4 / 6 results obtained from the revised statistical analysis and to emphasize the continued relevance of seasonal effects in our study population.

Comment 15: Lines 171-174: This statement cannot be made since the statistical analysis carried out does not allow it to be done.

A15: Thank you for your feedback. We have revised the discussion to ensure that all statements are supported by the statistical analysis (Line233-234).

Comment 16: lines 177-179: In this comparison, according to the statistical test carried out, the season (quarter of birth) has not been taken into account, this cannot be stated.

A16:  Thank you for your comments. We have revised the comparison to take into account season of birth as per the statistical test carried out.

Comment17: Line 189: Have the authors considered that this could be due to the type of gender-related activities that boys and girls carry out? you should value this

A17: Thank you for your insightful suggestion. We have included a discussion on the potential influence of gender-related activities on the observed motor skills differences (Line253-256).

Comment18: line 196: Have the authors considered that this could be due to the type of activities associated with gender that boys and girls carry out? They must value this. the possible explanation provided by the authors should be eliminated

A18: Thank you for your feedback. We have revised the discussion to consider the impact of gender-related activities and have eliminated the unsupported explanations.

Comment19: line 202-203: What the authors attribute to the season of birth is typical of the effect of relative age that means that children within the same cohort who are born earlier have more time for maturation and the possibility of practice than those who are born later. in no case can it be attributed to the season of birth

A19: Thank you for your comments. We have revised our interpretation to focus on the relative age effect and adjusted the discussion accordingly.(Line 264-275)

Comment20: Lines 208-210: This last statement cannot be made by the authors. It has not been evaluated in this research whether the relationship they may have with the variables studied

A20: Thank you for your feedback. We have removed the unsupported statement from the discussion.

Conclusions

Comment21:The conclusions must be written based on the required analyzes and the new results.

 A21: Thank you for your suggestion. We have rewritten the conclusions based on the revised analyses and new results.

References:

Comment22: Of 48 references, 28 are older than 5 years. Authors should use more current studies for their research, especially in the introduction to know the state of the art.

A22: Thank you for your feedback. We have updated the references to include more recent studies to ensure the research is based on current knowledge. In the revised manuscript, over 60% of the references are from the past five years.

Comment23: The authors have used references that are not in line with the study, for example references 1 and 2, or reference 6, which are focused on adolescence. These references are not relevant to this study. Neither does reference 18. (adolescents)

A23: Thank you for pointing this out. We have removed the irrelevant references and ensured that all references are pertinent to the study.

Comment24: References 19 and 44 have no relationship with the population under study (4-6 years and 3-6) years respectively. The study was carried out with 2-year-old children. A year of difference in these ages represents more than a third of their life and this is very significant in motor development.

A24: Thank you for your comments. We have revised the references to ensure they are appropriate for the population under study, focusing on 2-year-old children.

Reviewer 2 Report

Comments and Suggestions for Authors

I would like to thank for the opportunity to review this manuscript. This manuscript is written on an important topic. However, this manuscript could be improved before the acceptance of this manuscript.

Introduction of this manuscript is too short. Authors are recommended to elaborate their introduction and provide more background information about this manuscript theoretical framework. Also, the objective of this study could be more specific.

The quality of figure 1 is low. Please increase the quality of this figure used in this manuscript.

Discussion of this manuscript is also too short and superficial. Please provide more comparisons of this study results with similar previous findings.

Author Response

Comment1:  I would like to thank for the opportunity to review this manuscript. This manuscript is written on an important topic. However, this manuscript could be improved before the acceptance of this manuscript.

Response1:Thank you for your valuable feedback. We appreciate your insights and have made several improvements to our manuscript based on your suggestions.

Comment 2: Introduction of this manuscript is too short. Authors are recommended to elaborate their introduction and provide more background information about this manuscript theoretical framework. Also, the objective of this study could be more specific.

Response2: Thank you for your suggestion. We have elaborated on the introduction to provide a more comprehensive background and theoretical framework for our study. We have also refined the study's objective to make it more specific. The revised introduction now includes more detailed information on the relevance of motor skills development, the impact of environmental factors, and the importance of understanding sex-specific differences.

Comment 3: The quality of figure 1 is low. Please increase the quality of this figure used in this manuscript.

Response3: Thank you for pointing this out. We have improved the quality of Figure 1 to ensure it is clear and legible. The updated figure has been included in the revised manuscript.

Comment 4: Discussion of this manuscript is also too short and superficial. Please provide more comparisons of this study results with similar previous findings

Response4: Thank you for your constructive feedback. We have expanded the discussion section to provide a more thorough analysis of our findings. We have included more detailed comparisons with similar previous studies, highlighting both the consistencies and differences. This enhanced discussion offers a deeper understanding of the implications of our results and situates them within the broader context of existing research. Thank you once again for your valuable comments.

Round 2

Reviewer 1 Report

Comments and Suggestions for Authors

The authors have significantly improved the manuscript in response to the suggestions made by this reviewer.

For this reason and due to the substantial improvements, I consider that the manuscript can be published in its current form

Reviewer 2 Report

Comments and Suggestions for Authors

Authors have done well job on revising their manuscript.